# Gender Discrimination Insights in Romanian Accounting Organisations

**DOI:** 10.3390/ijerph20010797

**Published:** 2023-01-01

**Authors:** Widad Atena Faragalla, Adriana Tiron-Tudor, Liana Stanca, Delia Deliu

**Affiliations:** 1Faculty of Economics and Business Administration, University Babeş-Bolyai, 400591 Cluj-Napoca, Romania; 2Faculty of Economics and Business Administration, West University of Timişoara, 300115 Timişoara, Romania

**Keywords:** discrimination, women, accounting profession, perceptions, Romania

## Abstract

The paper investigated gender inequalities in the accounting profession in the specific context of an eastern European country, in the past heavily dominated by men, but now with a substantial number of women. Thus, we used a questionnaire survey explored the relationship between women’s perception of discrimination and institutional and individual characteristics. Institutional factors such as rewards practices, opportunities, and access to top positions in organisations influence women’s perception of gender discrimination. Concerning the intersectionality of individual characteristics interaction, our results revealed that women professionals with higher academic achievement and children are more likely to report discrimination. In contrast, women professionals with a higher-ranking position in organisations and those working in the public sector are less likely to report it compared with those from the private sector. These findings are of interest from a theoretical perspective to those who explore gender-related issues in general and in the case of accounting organisations. They are also helpful from a practical standpoint regarding the management of these accounting organisations, which should ensure gender-equitable policies for employees.

## 1. Introduction

Gender equality is not only a fundamental human right but also a necessary foundation for a sustainable world. Furthermore, it represents one of the fundamental values of the European Union (EU) and one of the 17 internationally accepted Sustainable Development Goals promoted by the United Nations (UN). The EU and the UN are dedicated to defending this right and promoting gender equality within the EU member states and across the world. However, some visible or invisible forms of inequalities exist in many regions, countries, sectors, and industries; thus, women continue to encounter barriers to equal treatment in the workplace. 

Women’s discrimination—or gender inequality—in society and especially in organisations and workplaces is a topic of fierce debate in modern society [1,2], especially in Western, developed countries. Discrimination in the workplace affects the professional career development of women [3] in terms of employment and advancing up the corporate ladder to management and executive positions [4]. A possible explanation refers to mentalities often based on stereotypes and biases [3,5]. Compared with men, women still tend to be employed less and work on average six hours longer per week than men (paid and unpaid) but have fewer paid hours. In addition, women take more career breaks and receive fewer and slower promotions [6]. 

Across all sectors and professions, different forms of gender discrimination exist [7], especially in professions traditionally dominated by men, such as accounting [8]. However, there is little evidence indicating whether gender bias continues to be an issue in professions where women’s representation has now substantially increased, such as the accounting profession. Historically, this profession has been dominated by men; however, it has recently grown to employ more women. Even so, getting hired for management positions remains difficult for women [9,10,11]. Most studies debating this issue are qualitative, focusing on European developed economies or regions other than Europe [12]; they reveal gender discrimination determinants such as age, childhood, marital status, education, position in an organisation, organisational practices, and societal and cultural stereotypes. 

Nevertheless, gender issues in accounting organisations have still been little researched [11,13,14]. Thus, this study closes a gap in the literature from two perspectives: first, through a quantitative approach based on Sever (2016) and Cohen et al. (2018), who examined the existence of discrimination in accounting organisations, and second, by exploring the little-explored Romanian context [3,15]. 

Romania’s emerging economy has a unique context concerning gender stereotypes and inequalities [16]. Romania is an Eastern European, former communist country, where the communist party promoted gender equality and women’s emancipation policies, and the myth of the “heroic working mother” was a central feature of Romanian political ideology [17]. However, even after the post-communist regime, in our days, Romanian culture is still essentially patriarchal, and women are perceived as less able than men to lead. In addition, childcare is still considered almost exclusively a woman’s responsibility and one of her significant tasks [18]. In Romania, even though there is a national regulation concerning gender equality (Law No. 202/2002), progress towards gender equality is slower than in other EU member states. In 2020, with a score of 54.4/100, Romania ranked in the lowest position in terms of gender equality [19]. 

Therefore, the purpose of this study was to investigate the perceived institutional and individual characteristics that induce the perception of gender discrimination in Romania based on the results of a survey of 130 professional women accountants. The accounting profession is noteworthy because the percentage of women in the profession is above 78%, making Romania the European country with the highest participation of women in accounting. However, there is still a lack of representation for women in the higher ranks of accounting organisations, despite the high number of women in the profession [20]. 

This study provides several unique contributions, such as the following. Firstly, the study contributes to the literature by exploring a complex topic because discrimination perceptions are subtle and subjective, difficult to define. Secondly, the study contributes to the existing theories by revealing factors that add to gender discrimination in a less-explored social context, i.e., that of a former communist country in Eastern Europe. Placed in the Romanian context, we explored a profession that now is numerically dominated by women, but who lack decent representation in top positions; to this, we added the collective mentality that preserves and still influences a great deal of the general perception of gender discrimination. 

The rest of this article is structured as follows: Section 2 presents the national and professional context of the research, followed by Section 3 and Section 4, a description of women’s discrimination based on the literature and the development of hypotheses. Section 5 describes the methodology, and Section 6 and Section 7 presents the results and discusses them. Finally, Section 8 contains practical implications, research limits, and future developments.

## 2. The Background of Women’s Discrimination in Romania

A longitudinal historical perspective before and after the decline of the communist regime facilitates a better understanding of the cultural legacy that influenced women’s role in society and workplaces in Romania. Today, the Romanian context differs from others for the following reasons. 

For more than 40 years, based on the Soviet model, the Romanian communist regime promoted official policies of gender equality, enabled equal access to education and employment, included women in the public arena and improved childcare systems, and extended maternity leave [18]. The downside was an aggressive anti-abortion policy, which included outlawing abortion and contraception, routine pregnancy tests for women, taxes on families with no children, and legal discrimination against childless people, as per the State Council Decree 770/1966 [21]. Another institutional norm was that women should be in charge of departments that have a large majority of women employees. In addition, women should be represented in leadership positions within the communist party, executive councils, and trade unions [17]. 

After the fall of communism in December 1989, as with all countries in Central and Eastern Europe (CEE), Romania made the transition from a planned economy to a market economy, and from a totalitarian regime to a democracy. Nevertheless, in that tumultuous period, promoting equal opportunities and treatment for men and women was not a priority for the government. Women were affected by this transformation, with lower wages, higher unemployment, and higher underemployment rates than men [22]. Employment rates for women, which were previously among the highest globally, declined, making CEE the only region in Europe where the integration of women into the labour market tended to decline rather than increase in recent decades. In addition, the CEE region has had a relatively poor work–life balance and a low part-time culture (below the EU average). Housekeeping and child-rearing in this region are still primarily seen as a woman’s responsibility [23]. 

Two explanations could account for the accentuated gender discrimination in transition economies. The first is the reverse effect of the communist legacies [23]. Secondly, the results could be traced back to the erosion of social support (childcare facilities, training opportunities), the change in social roles, and the exacerbation of negative stereotypes towards women during the democratisation process [22]. 

Thirty years after the collapse of socialism, Romania has 19.5 million inhabitants, 51.38% of which are women. However, in general (as shown in Figure 1), Romania ranks 26th according to the gender equality index [19]. Furthermore, the country’s gender gap in terms of access to paid employment is one of the largest in the EU.

Nevertheless, Romania is progressing towards gender equality, albeit at a slower pace than the other EU Member States [19]. One of the elements that holds back progress is the time spent on care activities that have increased over the past 20 years. Women spent more time caring for children, grandchildren, the elderly, or people with disabilities than they did in 2010. This gap is the most significant gender gap in the EU. In addition, the gender gap in the full-time employment rate between single women and men and couples with children is tremendous [19]. Opposite, a positive aspect is the distribution of income between women and men, which became more even between 2007 and 2018. The gender pay gap is currently 4% in Romania, the lowest in the EU [19]. 

Despite substantial changes in recent years, the participation of women in political life in Romania is still meagre [18]. In industrialised and less developed countries, including Romania, men in managerial positions in the economy far outnumber women [24]. Moreover, due to stereotype-based perceptions, women are still underrepresented in decision-making positions at all levels, including in industries and professions where women are numerically dominant. For example, the proportion of women on the executive boards of the largest listed companies is low, although a considerable number of women are trained and qualified in business, administration, and law [19]. In addition, the proportion of women in large, listed companies’ executive boards fell from 13% to 10% between 2005 and 2017 [6]. In this sense, Coman (2015) pointed out that different personality traits between men and women prevent women from advancing to top management positions [24]. The characteristics of cultural organisations may offer adequate support to contribute to a better understanding of Romania’s cultural context. The national culture cannot be changed [25], and influences the corporate culture [26]. An open and accessible corporate culture cannot create an effective self-governing organisation if the national culture is quite individualistic and has little power over mentalities [25,26]. 

## 3. Theoretical Framework

This study is based on the gendered and gendering organisations theories [27,28], which connect the structure of the organisation with the people in it. These theories emphasise that the power-based processes that reproduce the gendered structure of organisations are also helpful to and involved in creating gender components of individual identity, meaning that wealth, power, and access to better opportunities are unequal between genders. Acker (1992a, 1992b) stressed that the term ‘gendered’ refers to the processes through which certain jobs privilege masculine constructs and gendering [27,28], and pointed to various discursive strategies to illustrate the fluid and socio-historical dynamics of the unveiling of gender norms and practices. Even if gender discrimination is a phenomenon that is difficult to quantify, it can be characterised by certain criteria such as power, income, privileges, education, and membership in certain social categories [29,30]. 

The theory of gendered organisation states that there are normative expectations in every workplace that gives one gender an advantage over the others. This status is reinforced by the gender assumptions of the universal worker, who is supposed to be incorporeal, but which is actually modelled after a man’s body [31]. With this de facto adopted, it is difficult to view organisations as gender-neutral systems influenced by the employee profile, rather than contexts and platforms in which gender attributes are created and reproduced. Acker’s approach also underscores how gender and other dimensions of inequality are embedded, even in seemingly formal, transparent policies and practices, such as job evaluation, guidelines, or job descriptions that otherwise appear gender-neutral. It can be argued that an organisation is inherently gender-specific when it is defined, conceptualised, and structured in terms of a distinction between masculinity and femininity, and this will inevitably reproduce gender differences [29,30]. 

These theories systematise and render visible all gender-related questions such as bias, discrimination, and inequality of opportunity as well as the distribution of privileges [16] and offer a platform for an open discussion on how a balanced and gender-specific organisation should look. The literature suggests that women’s retention in higher echelons of organisations has stalled; women are marginalised at this level, which in turn results in an underrepresentation of women. 

The accounting profession is already gendered [31]; theory on gendered organisations and institutions might be the key to explaining the gendered impact and how inequalities emerge in the accounting profession. Specific invisible but natural discriminatory obstacles exist due to implicit prejudice based on ethnicity, age, sex, caste, family, environment, and political and religious factors. These often prevent women from rising to positions of power and responsibility within society and organisations, but this entire realm is a debatable issue in many sectors. For example, gender discrimination in workplaces involves discriminatory treatment based on subjective criteria such as gender bias in the processes of selection, compensation, promotion, professional training, and the recognition of professional merits [3,32].

According to historical evidence, women have experienced gender discrimination far more than men [15,33]. For this reason, understanding women’s overall perceptions is essential since more organisations are intensifying their efforts to build a workplace wherein inequality is not an issue [34]. The perception of discrimination at work and inequality can become one’s daily reality [5], regardless of whether it is wholly accurate, and will have substantial effects on behaviours and work patterns. If a woman believes that there is gender discrimination in her organisation, she may no longer be as motivated or interested in promotion, rewards, and peer recognition. This situation may instead lead to creating a pattern in her behaviour that will only perpetuate the issue [5,35,36].

Although considered illegal, discriminatory practices are still prevalent in most countries and industries. The literature captures aspects of this issue by using ‘glass’ metaphors, such as ‘glass ceiling’ [37], ‘glass labyrinth’ [38], ‘glass cliff’ [39], ‘glass wall,’, or ‘glass escalator’ [40]. Others refer to gender discrimination using ‘maternal wall’ or the ‘sticky floor’ metaphors [41]. In the 1980s, the ‘glass ceiling’ phenomenon described an invisible or artificial barrier to women’s career advancement and organisations’ failure to promote women to top leadership roles [33,37]. A role congruity theory of prejudice toward women leaders was proposed [42] to explain the perceived incongruity between women’s gender role and leadership roles.

Two decades earlier, Eagly and Carli (2007) argued that the ‘glass ceiling’ metaphor had outlived its usefulness [38]. This context leads managers to overlook interventions that would attack the root of the problem. Thus, a labyrinth is a more fitting image to help organisations understand and address the obstacles to women’s progress. Traversing a maze requires perseverance, an awareness of one’s progress, and a careful analysis of the puzzles that lie before us. There are routes to the centre, but they are full of unexpected and unexpected twists and turns. The pressures of intense parenting and the increasing demands of most high-level careers have left women with very little time to meet with colleagues and build professional networks, that is, to accumulate the social capital necessary for leaders who want career progression. 

Women suffer from unequal gender representation at all levels, not just at the top, as influenced by institutional factors within organisations as well as social and cultural pressures [34]. Glass walls prove to be the initial obstacles to women in the workplace, ahead of the ‘glass ceiling.’ This issue occurs because women are often relegated to roles that do not require the required experience to be selected for top management jobs. Women are concentrated in particular roles and limited to specific management functions to indicate the ‘glass walls’ phenomenon, which constitutes occupational segregation by gender [43].

Another issue mentioned in the literature refers to gender stereotypes, which are widely held beliefs about the characteristics and behaviours of women and men [44,45]. Appetite for risks, competitiveness, and impartiality are associated with men, while patience, care, and kindness are associated with women. In addition, the mentality and stereotypical thinking of those men inherently position them as providers for their families or de facto managers in the workplace [33]. On the other hand, women are assumed as caretakers of the household and children, which has led to the perception that women are less committed to their careers and that they do not have the necessary traits and skills to lead [3,33]. Thus, successful managers possess characteristics, attitudes, and temperaments more commonly ascribed to men in general than to women in general [45]. 

Even though women make up more than half of the overall accounting workforce, they represent only 5% of the overall partners, CEOs, and senior managers of the top Fortune 500 companies [46]. In Europe, women make up almost two-thirds (62.8%) of the legal and accounting workforce [6]. Since more than half of these accountants are women, the gender pay gap and under-representation of women at higher levels are common issues [16,47]. 

In some countries, women represent a majority in this profession, such as Romania, which has the highest proportion of women in this profession (77.9%) out of all European countries [20]. However, throughout Europe, vertical segregation remains; that is, women are still unable to break the ‘glass ceiling’ to achieve equal access to top positions [48]. 

Understanding organisational practices and processes is central to explaining gender issues, especially in the accounting profession [49,50,51,52,53]. Since the accounting profession is rigid, and subject to rules that leave no room for flexibility, gender issues are inevitable and deserve to be studied [11,13]. 

To conclude, women’s discrimination would be more accurately described as a labyrinth with multiple obstacles that build up over time [8], thereby limiting the upward trajectory of women accounting professionals. Moreover, the labyrinth illustration offers a more delicate and intricate metaphor, unlike the others mentioned in the literature. This labyrinth is full of obstacles suggesting that women are confronted with several challenges throughout their working lives. Furthermore, the labyrinth metaphor explains that progression is very challenging, but possible. However, the labyrinth metaphor has been scarcely explored in the accounting profession, indicating the need to do so. For this reason, the following section extracts relevant factors from the literature that might influence gender discrimination in the specific context of Romanian accounting organisations. We grouped these issues into two primary components of the labyrinth: institutional practices and individual characteristics.

## 4. Development of Hypotheses 

### 4.1. Institutional Factors and Gender Discrimination

In an organisation, the processes of selection, compensation, promotion, mentoring, and networking are the areas wherein gender discrimination might be more evident. Even if the pay gap is mentioned in the literature as evidence of gender discrimination, this element was not considered relevant in the Romanian case with the lowest pay gap in Europe. In addition, professional training was not analysed because it is mandatory to continue professional development in the accounting profession, which includes a specific minimum number of hours worked. 

In many cases, people’s mentalities and upbringings cause unconscious biases that may cause men to act in a discriminatory manner [54]. By nature, employers are more inclined to choose employees with the same traits, resulting in an unconscious bias in men situated in higher ranks regarding the selection, promotion, or compensation of women in similar positions [5,13]. The gendered nature of the accounting profession and stereotypes of accountants and managers or people in positions of power have been patterned after stereotypically masculine characteristics. When a woman attempts to enter or advance in a position of power, she may try to break such stereotypes, sometimes trying to display masculine traits to succeed. In many cases, she can suffer from backlash for such attempts by not being hired [3]. Based on the abovementioned, the first two hypotheses were proposed: 

**H1.** 
*Women accounting professionals who consider there is unfavourable differential treatment between them and their men counterparts regarding performance evaluations and the outcome thereof (e.g., selection, promotions) are more likely to report a glass ceiling in their organisations.*


**H2.** 
*Women accounting professionals who perceive there is unfavourable treatment between them and their men counterparts regarding compensation and rewards are more likely to report glass ceilings in their organisations.*


Specific organisational policies and practices which are systematically applied—such as preferential treatment for men concerning mentoring, networking opportunities, and the availability of better and more high-profile job assignments—are also responsible for gender discrimination [3,8,54]. Mentoring implies some degree of grooming for a job that involves access to information otherwise restricted to others [55]. In addition, this practice introduces a new manager to the right social circles, opening up opportunities to network. In other words, mentoring refers to sponsoring mentored employees, offering them all the information needed to succeed. However, evidence suggests that most women succeed in management positions only with the help of a mentor, and many of them have difficulties in securing a mentorship [54]. Furthermore, as not many women are in the higher ranks of organisations, this perpetuates this phenomenon since there are few advocates for women’s advancement in existing hierarchies [11]. Moreover, under these conditions, women must often seek out men for mentoring opportunities, which is another situation in which multiple barriers can exist, such as a lack of interest in men mentors grooming women in the same way they would groom men [3]. Thus, in some organisations, women may feel excluded by not being included in certain circles from which they would obtain access to privileged information or access a mentoring system. This issue becomes clear when women marry and/or have children due to their additional marital and or motherhood-related duties [56]. However, women lack the support of a network and are less likely to socialise with men and clients over subjects of stereotypically masculine interest, making it harder for women to form such relationships [5,13]. Due to this exclusion from such networks, women are also left to disseminate vital information [34]. As a result, women have perceived that their work is less valuable and not as worthy as that of men [3,47]. Based on the abovementioned, the third hypothesis was developed: 

**H3.** 
*Women accounting professionals who consider that women employees do not have the same opportunities for professional development as men are more inclined to report glass ceilings within their organisations.*


Another important aspect regarding career advancement is related to obtaining high-profile job assignments. Men are sought after to take these kinds of ‘make or break’ career assignments, whereas women must request them. In gendered organisations, high-profile assignments are critical for women’s career advancement since they can help women to advance their careers [8,55]. According to cultural stereotypes, men in higher ranks of organisations are often inclined to form bonds with other men rather than women. Thus, they are more inclined to promote and accept men than women in the higher ranks of organisations. This lack of support from men leaders results in fewer promotions of women in top management positions [47]. Through this bonding, a network is formed that often helps men be noticed, giving them more advantages and leverage for promotion. In this way, such informal practices from which women are left out would greatly help in their quest to advance their careers [3,47]. Moreover, in Romania, the proportion of women on the executive boards of the largest listed companies is low [57]. As a consequence, the fourth hypothesis was developed as follows: 

**H4.** 
*Women accounting professionals who consider that women employees do not have the same level of support, are not mentored at the same level, and are not included in certain men-dominated networks compared with their men peers are more likely to report glass ceilings in their organisations.*


### 4.2. Individual Characteristics and Gender Discrimination

Individual characteristics encompass many aspects, including gender, marital status, motherhood, work experience, company rank, age, and education level, and different sectors allow differences between individuals to be distinguished. In addition, the literature mentions others, such as race, religion, and colour, but these are not relevant to this paper. 

Married women or those in a stable relationship with a man are less inclined to support efforts to reduce gender inequality [58]. These women tend to accept views from men, internalise them, and change the views they had up until that point. They may have fewer egalitarian views and tend to find other reasons as to why women do not make it to the top in the business world, such as individual lack of capabilities, motivation, and abilities [3,7,59]. On the other hand, single women (unmarried, divorced, widowed), especially those who are pregnant, are more likely to attribute the difficulty of climbing the hierarchical ladder to reasons such as access to employment, promotion, rewards, and fewer opportunities for networking and mentoring [10]. These arguments sustain our fifth hypothesis: 

**H5.** 
*Women accounting professionals who are not married or in a stable relationship are more likely to report gender discrimination in their organisations than those who are married.*


Furthermore, motherhood bias occurs when colleagues view mothers—or pregnant women—as less competent and less committed to their jobs [60]. Men with children experience more career success than men without children, while women with children tend to experience less career success than women without. The manifestation of this inequality might take different forms, coming from hiring committees, colleagues, and individuals conducting performance evaluations. In addition, mothers may be overlooked for challenging assignments or promotions because of their assumed lack of time or desire [61]. Sometimes these women are directly told they should be home with their children. Conversely, based on findings in the literature, it seems that men who have children have more success in their careers, whereas women with children experience less success [8,9,58]. In Romania, more women (65%) than men (50%) are involved in caring for or educating their children or grandchildren at least several times each week [6]. Age and gender mistreatment at different life stages are two factors that converge. Women’s working lives are characterised by high rates of mistreatment throughout their careers, in a way that men’s are not [62]. For example, their employment chances decrease with advancement in age and after breaks from work to raise children [63]. Based on the literature discussed, the sixth hypothesis was formulated as follows: 

**H6.** 
*Women accounting professionals with children are more inclined to perceive and experience glass ceilings in their organisations.*


Women who are highly ranked in their organisations experience discrimination (proven by numerous studies) since they have to break through barriers to make it to the top. However, despite this fact, they are less inclined to recognise that their organisations are gendered by nature, that gender discrimination exists, and that they must face multiple obstacles in their careers that men do not have to face [64]. Such women may be more inclined to report the existence of gender discrimination since they have experienced it. This situation is an issue since these women are in a unique position to change their mentalities and wrong practices, but if they believe that they should not speak up, they are only perpetuating gender discrimination and not dismantling it [7]. Chow et al. (2002) found that higher-ranking accounting professionals who tend to think highly about their organisations and see the system as fair are less likely to report gender discrimination in their organisations [65]. Thus, our seventh hypothesis was proposed as follows: 

**H7.** 
*Higher ranking women accounting professionals (senior executives/directors and partners) are less likely to report the existence of gender discrimination than those who are part of the lower ranks of organisations.*


With a higher rank generally comes more years of work experience and thus seniority. The seniority leads to more time to experience gender discrimination and to observe certain practices that are unequal for women [31]. Additionally, there are many instances when even though they have been discriminated against throughout their career, women do not and will not acknowledge it out of a concern that they might invalidate their success. Following this idea, the eighth hypothesis was proposed as follows: 

**H8.** 
*Experienced senior women accounting professionals are more likely to report gender discrimination than those who are entry-level professionals.*


In addition, this phenomenon is strongly connected to education. Women with higher education are more informed; thus, they know their rights better and are more likely to speak up. Concerning education, according to accounting professional national regulations, only persons with a higher education diploma may be accepted as accountant experts or auditors, whereas others may be accepted as only licensed accountants. Both might begin work as independent legal workers, but only an expert accountant might open an accounting company. Thus, a ninth hypothesis was proposed as follows: 

**H9.** 
*Women accounting professionals who hold higher studies certifications are more likely to report gender discrimination than those who are less educated.*


The collective mentality formed just one or two generations ago encourages people to not complain about the possible inequalities in organisations and not to act upon them. Instead, our ancestors were taught to act as if it is a man’s world and that it is normal for women not to be well represented in higher ranks. Thus, women of greater age are more likely to perceive gender discrimination, but are not necessarily keen or aware enough to openly address it [24]. This situation could stem from the fact that those subscribing to these beliefs are uninformed. Perhaps in their youth, gender discrimination was not recognised, and maybe they are not even aware that what they are experiencing is gender discrimination. This gap is strongly connected to education as well, since women with higher degrees are often assumed to be more informed. Thus, they know their rights better than uneducated women, and they dare to speak up since the generation they are part of has more access to better resources. To our knowledge, this correlation between age and education and gender discrimination has not been made in another study. In consequence, the tenth hypothesis was proposed as follows: 

**H10.** 
*Old-aged women accounting professionals are less likely to report glass ceilings than younger accounting professionals, much owing to generational differences in up-bringing.*


Another cultural legacy refers to the inherited idea that jobs in the public sector are typically more secure than in private enterprises, and more vulnerable to market fluctuations. In addition, public sector jobs tend to offer a better work–life balance, more organisational commitment and support, and less burnout than the private sector [66]. As a logical assumption, gender discrimination is more likely to be present in a private accounting organisation than in a public one [66]. The statistical data confirm that gender gaps are extensive among women and men working in the private sector (75% and 47%) [6]. Women with young children often choose job security and family-friendly work hours available in the public sector, typically feminised jobs over higher wages in private-sector positions, which are more vulnerable to economic volatility. However, such ‘choices’ may not be entirely voluntary. Instead, they may occur due to prohibitively lengthy work hours, barriers to entry into private, professional sectors, a lack of supportive family arrangements, or inaccessible or insufficient childcare facilities. All these arguments are at the foundation of the eleventh hypothesis: 

**H11.** 
*Women accounting professionals in public accounting are less likely to report the existence of glass ceilings within their organisations than women accounting professionals in private accounting.*


This study used these categories for quantitative research but acknowledges that all of them are socially constructed, imperfect, and not fixed in time. In addition, the presumption is that these categories do not act separately but rather in a synergistic way. Thus, one can better understand the dynamics of gender, age, education, rank, experience, and life events [67]. This study embraced an intersectional approach, aiming to capture the interactions between categories [68], focusing on particular intersections [69] since specific categories intersect more than others. However, even within tightly defined intersectional positions, substantial individual heterogeneity will always remain.

## 5. Research Methodology

The methodology consists of a set of steps performed to establish the factors that influence gender discrimination in a pre-established context. The questionnaire’s content was established based on the relevant literature debates. The study continues with the empirical part consisting of an econometric analysis of the collected responses to the questionnaire. Each step performed is explained below.

### 5.1. Questionnaire Development

First, the exploratory study was based on the literature review findings and interviews with five social sciences researchers who contributed to the set-up of elements included in the questionnaire. Second, the questions were formulated to transparently address all aspects determined in the literature about gender discrimination. 

Our option for a dichotomist scale of answers was to avoid neutral responses or use a multi-criteria scale to create confusion for the respondents [70,71]. Respondents who do not want to reveal their opinion use the neutral response in general. Thus, to measure the gender discrimination perceptions, the dichotomist scale is better because the possibility to choose a neutral answer is eliminated. In this way, we can obtain a more critical assessment closer to reality [70,71]. Based on the literature review, gender discrimination is a simple single-faceted construct, and it is impossible to create many different items that measure the same underlying construct [3]. Moreover, Rossiter (2002) argued that “a singular concrete object to be rated in terms of a concrete attribute needs only a single-item scale” [72] (p. 331). We consider single-item measures suitable for our study despite their low level of reliability and validity. The main reasons, which balanced our rationale in favour of it, are because they minimise the respondent burden, reduce criterion contamination, and increase face validity, as, for example, mentioned by Fisher et al. (2015) [73]. Moreover, our study goal was to understand the general nature of women’s discrimination and to obtain an overall feeling, judgment, or impression about women’s discrimination in an accounting context. 

Next, the factors that influence gender discrimination were transposed into measurable statements systematically included in the questionnaire content. 

The first group of factors includes perceptions related to promotion, rewards, opportunities, and access to top management positions in organisations. In other words, factors were included if they address a difference in behaviour inside organisations between men and women. The following independent variables describe these institutional factors, as shown in Table 1.

The questions related to H1, H2, H3, and H4 were stated in a neutral frame (women are treated differently, not women are treated worse) in order to prevent the consequences of leading the respondent [74,75]. 

The second group of questions refers to individual characteristics of women described by the following independent variables, as shown in Table 2.

Married women accountants are less likely to report gender discrimination, although children are more likely to report it. In addition, these two variables were intended to capture the beliefs of single women and unmarried women with children, along with women who are divorced or widowed with children. 

Higher-ranking professional accountants are less likely to report gender discrimination even if they experience it, whereas lower-ranking women accountants are more likely to report it. In addition, women accounting professionals with seniority (more years of work experience) will be more likely to report gender discrimination, whereas those with less work experience will likely not file such reports. 

We expect that women accounting professionals who hold higher certifications would be more likely to report gender discrimination and that older women accounting professionals would be less likely to report gender discrimination than younger accounting professionals. Finally, we expect that women accounting professionals working in the public sector would be less likely to report gender discrimination than those in private-sector organisations. 

To obtain a general view of gender balance at entry-level, middle, and top management, we formulated three questions: the man/woman ratio at each level, with the possible answers: balanced, more men, more women. 

Based on the operationalisation of theoretical constructs, before conducting the leading research, the questionnaire was discussed with seven practitioners and academic respondents to check the clarity of questions, and after taking into consideration their inputs, the questionnaire was finished. 

### 5.2. Questionnaire Validation

To test the questionnaire’s reliability and validity, we collected the answers from 30 respondents working in the field of accounting. Reliability refers to the measure in which the questionnaire can produce consistent results, and the validity reveals the level of adequacy of the given questionnaire for the studied area; the number of items is adequate [76].

We refer to the internal consistency of the elements, testing whether they correlate well or not. For this case, Cronbach’s alpha version for dichotomous items Kuder–Richardson test KR20 was used [77], allowing us to measure the dichotomous fidelity data, and KR21 indicates if the items are reliable when measuring the same construct, and also the accuracy, stability, and coherence of the test results. Since the KR20 coefficient provides minimum reliability estimates and the difficulty of the items in this section is heterogeneous [78], all items were upheld for analysis. The difficulty of the knowledge items varied from 30% to 97%, averaging 59%. Gender discrimination measures the degree of correspondence between the success in each item and the whole set of items and can be computed using a biserial point correlation [79]. The correlation values must be above 0.30 for items to be considered sufficiently discriminating [79]. Although the scores obtained may suggest a moderate discriminatory capacity, we demonstrate the questionnaire’s capacity to distinguish gender discrimination. 

The Kuder–Richardson test (KR20 = 0.797 and KR21 = 0.715) confirmed that there is no need for the reorganisation of the instrument used to analyse gender discrimination. It confirmed the same regarding the reliability of the items forming the instrument employed for measuring the presence/absence of gender discrimination. The values obtained are significant and confirm an excellent internal consistency of the items in the questionnaire; the instrument is adequate for the purpose for which it was built, so one can assert with certainty that the test is one-dimensional. The values and significance of the discriminating coefficient in this study revealed an appropriate level of reliability, so the questionnaire items tend to form a unit by themselves. 

We continue the analysis of Inter-rater reliability (and also Intra-rater reliability) for qualitative (categorical) items (across respondents), using Cohen’s kappa [80] to measure the agreement between the respondents subtracting out agreement due to chance. The Cohen’s kappa coefficient (Kappa = 0.80; Z = 12.404; *p*-value = 0.002) values are substantial according to Marston (2010) [80]. Moreover, the respondents’ answers were different. They did not give the same value to the items included in the questionnaire, so the study’s scale allows a critical assessment of gender discrimination. Based on the reliability measures concerning internal consistency and interpreted reliability, we can state that the construct is also valid.

We conducted an exploratory factor analysis (EFA) to identify the main dimensions created through factor analysis using Varimax rotation. EFA assumes that a small number of latent constructs are responsible for the correlations between large numbers of observed variables [81]. In order to select the components of specific factors, a measuring model whose value is greater than 0.6 is considered reliable [81]. The analysis started with the determination of the Kaiser–Meyer–Olkin test, which was 0.647, *p*-value < 0.001, which demonstrates that it is advisable to perform a factor analysis. At this stage, we decided if the instrument used has an average degree of confidence or consistency, so that its results are the same over time and can be used in scenarios such as those discussed in this article. This proves that the sample used is sufficient for the study and as a result, the variability of data is caused by the instrument created in Bartlett’s Test of Sphericity, assuming all correlation coefficients are not quite far from zero. Bartlett’s Test Sphericity (412,791, Sig = 0.000) is small enough to reject the hypothesis, according to which the variables are not correlated.

### 5.3. Data Collection 

To collect survey responses, we sent out 1500 emails to women professional accountants from all major cities in Romania (Cluj-Napoca, Brasov, Iasi, Bucharest, Timisoara, Constanta), using the available public contact information. The request was sent by email and included a description of the purpose of the study and the link to the web-based survey (using the isondaje.ro tool). After sending the initial round of emails, this team sent a one-week reminder email. In total, 130 responses arrived from women participants, which were the only ones considered. The response rate was 8.67%, similar to the rates obtained by Cohen et al. (2018)—6.1%; Dalton et al. (2014)—8.1%; and Anderson and Lillis (2011)—5% [3,5,82]. 

The focus of this study was to analyse women’s sense of gender and their social backgrounds and trajectory within the accounting field, since discrimination has historically been viewed as an issue experienced mainly by women [15,35], especially in the accounting profession [49,50,51,52,53]. 

We, therefore, do not claim that the “findings” are objective or, indeed, generalisable, but rather offer insights into the complex nature of women’s experiences of gender discrimination in a former communist country.

To assess whether the non-response bias is problematic in this study, this team compared the responses of early and late respondents (i.e., the first 25% and the last 25%, respectively), as Armstrong and Overton (1977) recommended [83]. Statistical significance was estimated with Chi-square tests. The *p*-value > 0.05 obtained, which was statistically not significant, allowed us to conclude that there are no significant differences between early and late respondents for the variables used in this study. The result obtained is expected because the remainder of the replies sent came fewer than 15 days after the initial email, signifying that the answers received in the second downstream do not come from non-respondent characteristics. In addition, the respondents acted voluntarily, without any constraints or connection with the authors. 

After collecting the responses, coding, and checking the answers obtained from the questionnaires, the data were analysed with SPSS 13 (IBM Corporation—IBM SPSS Statistics, New York, NY, USA).

### 5.4. Econometric Analysis

The statistical analysis of the data investigates the following issues: The main characteristics of the respondents offer a description of the sample using frequencies.The relationship between variables with Spearman’s correlation coefficient reflects associations, not causal relationships. Then the relationships between variables were validated using the chi-square test for independence.The logistic regression to find the independent variables (institutional and individual characteristics) that influence the binary dependent variable of gender discrimination [84]. The method selected for the logistic regression was the Enter method; the factorial variables were simultaneously tested. The results model estimates the probability that a factor is present in respect of different explanatory variables and its 95% CI was estimated for each factor, a *p*-value < 0.05 (two-sided) was considered to be statistically significant. In general, the logistic regression model showed that the effect of a covariate on the chance of “success” is linear on the log-odds scale, or multiplicative on the odds scale. We used the B coefficient as a linear regression model. The statistic -2logL represents a badness-of-fit measure. Large numbers mean a poor fit of the model regarding the data. The statistic chi-square was used to test whether a variable reduces regarding its badness-of-fit measure [84]. A significant chi-square revealed that the independent variable is a very good predictor in this model.Through the Receiver Operating Characteristic (ROC) curve procedure, we tested if the attributes taken into the study are predictive for the chosen model to detect gender discrimination perception. We achieved the evaluation of the performance model through the receiver operating characteristic curve [85]. The ROC analysis was used to check the power of gender discrimination that the survey had. The result of this test offers us the possibility to conclude whether the gender discrimination test is a discriminative model or not. Thus, by using the ROC curve procedure, the attributes taken into the study were tested if there are predictive for the chosen model.

## 6. Results

### 6.1. Descriptive Results

The sample of women respondents’ individual characteristics are presented as follows in Table 3.

Most of the respondents are working in the private sector, are married, have children, have higher education, and have experience with or without a management position in the company where they are working. The women’s perceptions towards gender discrimination are quite balanced, but in favour of not perceiving gender discrimination at the workplace. However, the gender balance at the different organisational levels presented below showed contrasting results, as shown in Table 4.

The entry-level values reflect the accounting profession’s inclination towards feminisation; only 11% of respondents indicated that more men than women are entering the accounting profession. For gender balance at the middle management level, the results indicated more women, similar to entry-level statistics. However, the situation is changing at the top management level, where men are very well represented compared with women. For the Romanian context, the results indicated that gender balance is more than achieved for women at the entry “sticky floor” level [41], but not so much at the top management level. From a masculine perspective, there is an 11% percentage at entry-level and 36% in top management positions, and thus the “glass escalator” phenomenon [40] is clear, revealing at the same time that men are on a fast track to top management positions when entering the Romanian accounting profession, which is dominated by women in reality.

### 6.2. Correlations and Associations

The statistical analysis continues with the application of the Spearman test to identify the correlation between gender discrimination and the variables studied (Table 5). For the Spearman correlation, the following intervals were considered: (0−0.19) very weak, (0.20−0.39) weak, (0.4−0.59) moderate, (0.6−0.79) strong, and (0.8–1) very strong.

Based on Spearman correlations values, gender discrimination is correlated weakly even in the case of persons who are unwilling to declare gender discrimination in their organisations. 

Gender discrimination was positively weakly associated with the following variables: sector, education, children, and married, especially amongst the respondents affirming that they had not experienced gender discrimination in their organisations. There was a weak negative association of gender discrimination with promotion and rewards and access to top management positions. This means that these three factors are not determinants of gender discrimination and that women do not perceive these factors as things that would cause them to state that there is gender discrimination in their organisations. A weak association of gender discrimination existed with rank and opportunities (+) and age (−), implying that women in senior positions are less inclined to admit gender discrimination in their organisations. In many cases, these women are older and have more experience as well.

In addition, there was a robust association between age and seniority since older women often have more experience than their younger counterparts. Furthermore, age was moderately correlated with rank (−), showing that older people experience gender discrimination, but they manage to advance to top positions and overcome such invisible barriers by working hard. They regard the process of admitting that gender discrimination in their organisations would minimise their work and invalidate their success. Thus, they prefer not to be distracted from the achievement of advancing up the hierarchical ladder. 

Between seniority and having children and being married, there was a weak correlation, implying that women with experience in the accounting profession have children and are married and are less likely to report gender discrimination. 

Furthermore, here the collective local (Romanian) mentality interferes [26,48]. That is, hard work to make it to the top is perceived as both a benefit and a badge of honour, transforming gender discrimination into a competitive advantage. As a result, older women are less inclined to admit that gender discrimination exists than the younger generations, who can openly admit that this phenomenon is real and represents an obstacle in their careers. Older people, who from this analysis are the ones who have the most jobs in higher-ranking positions, do perceive gender discrimination, but they are reluctant to admit it. Instead, they believe they have reached their positions by working hard, not wishing to invalidate their success. 

Promotion and rewards are moderately correlated, and biases are weakly correlated with opportunities and top management. This means that even if women do not perceive promotion, rewards, and access to top management positions as determinants, they think that these three factors are connected and can influence each other. Opportunity was weakly correlated with top management positions, so women who believe men have more professional opportunities than women are more likely to report gender discrimination. Another strong correlation existed between education and the private sector; generally, women with higher studies work in the private sector. 

In order to verify the results obtained by Spearman, this study considered the use of the Chi-square associations test. The values of the Pearson Chi-Square test among all considered variables and their significance levels are detailed in Table 6. All associated *p*-values were well below the traditional cut-off value of 0.05, reflecting relationships between gender discrimination and the studied variables (Table 6).

According to the Pearson chi-square test’s results, women who have children declared that they have experienced gender discrimination in their organisations, which is linked to the level of education since they have superior degrees and higher levels of education. 

Regarding seniority, there is a positive association between gender discrimination with women who have less than 18–24 years of work experience, meaning that they have experienced gender discrimination more often than those with between 18–24 years of experience. One explanation of this finding is that people with 18–24 years of experience are used to dealing with this phenomenon, so they become resistant to it. Thus, they do not perceive it as frustrating, as do people who have less experience. In addition, another explanation could be that they still have communist influence in their mentality since they were developing young adults when Romania was a communist country. Therefore, they are not accustomed to a culture of standing up for themselves and their rights. 

The sector variable was also associated with gender discrimination. However, it seems that women working in the public sector are less likely to experience gender discrimination than those working in private companies. 

The association between rank and gender discrimination was positive, proving the hypothesis that women in high ranks are inclined to report gender discrimination as the literature is highlighting [34]. The results showed that women in top positions admit that they experienced gender discrimination, but they are not inclined to declare that it has affected their careers or report it. On the other hand, women in middle management positions were more inclined to report it and stated that this phenomenon does affect their careers. 

Promotion and rewards are associated with gender discrimination and refer to women’s belief that men are preferred to women in their organisation when it comes to promotion or rewards. Therefore, the existence of promotion and reward bias implies the existence of gender discrimination; thus, promotion bias influences gender discrimination and not the other way around. 

The Pearson’s chi-squared test confirmed the existence of gender discrimination in accounting organisations in Romania. The results showed that the women accounting professionals who have seniority in the profession—thus implying that they are potentially more likely to have higher ranks in their organisations—are reticent in stating that they have experienced gender discrimination in their organisations or have confronted this phenomenon for many years. On the other hand, younger people are more open and not held back by anything. Moreover, women working in the public sector are less likely to experience gender discrimination than those in the private sector. 

The existence of associations, revealed by Pearson’s chi-squared test, validated the results of the Spearman correlation test. In this context, this team applied the logistic regression test, which is presented in the following section.

### 6.3. Factors That Influence Gender Discrimination

Logistic regression was used to determine which of the variables was a direct determinant of gender discrimination. 

The results of the logistic regression test are listed in Table 7. The results of the Chi-squared test (chi-squared = 58.547; *p* < 0.001) and of the verisimilitude rate 2 log-likelihood (2LL) in step 1 compared with step 0 confirmed that the analysed variables influence gender discrimination and allowed us to accept the variables that were chosen as determinants of the gender discrimination (Cox and Snell R2 = 0.363; Nagelkerke R2 = 0.487). For any item for which we have a directional hypothesis, we report one-tailed *p*-values. Hosmersi–Lemeshow test’s results suggest that the variables influence the studied phenomena. The odds ratio indicates for each variable the probability to be predictive of gender discrimination. The values of predictor variables were close to 1, and all the Variance Inflation Factor (VIF) values were lower than the ten criteria, indicating that there were no multicollinearity variable predictor issues that might affect the regression model parameters [84]. We report two-tailed *p*-values for all other items, and the model has the following elements and values.

The logistic regression model produces a predictor—that is, a weighted combination of explanatory variables or covariates—for gender discrimination. Our gender discrimination model reflects the direct effects of the seven independent variables that become significantly amplified by each other.
*Gender discrimination* = −4.185 + (−2.356) × *REWARD* + 3.544 × *OPPORTUNITY* + 1.142 × *TOP* + (−1.076) × *CHILDREN* + 1.932 × *RANK* + 1.436 × *EDUCATION* + 1.108 × *SECTOR*

The logistic regression results indicated that institutional and individual characteristics significantly influence women’s perceptions of gender discrimination in the workplace. First, at the organisation level, variables such as opportunities for networking and mentoring, organisation practices concerning rewards, and top management accession influence it, following Hull and Umansky (1997), Lupu and Empson 2015, and Cohen et al. (2018) [3,8,54]. Next, the following individual characteristics, such as rank occupied in an organisation from the public or private sector, education level, and motherhood, significantly influence women’s perceptions concerning gender inequalities that exist in organisations. 

In summary, this study showed that gender discrimination is perceived by Romanian women accounting professionals in different ways depending on their situations. This sample included perceptions of women with children (*p*-value 0.0457); working in the private sector (*p*-value 0.0417); with higher education (*p*-value 0.0322); aware of workplace practices concerning opportunities (*p*-value 0.0416); rewards (*p*-value 0.0229) and with a superior position in organisations (*p*-value 0.0097). A value of the Odds ratio bigger than 1 suggests a strong presence of gender discrimination for the items because the presence has better odds than absence. The model obtained via the logistic regression recognises rank, top, opportunity, reward, children, education, and sector from all the variables tested (children, education, seniority, rank, married, sector, rewards, promotion, opportunities, top) as the ones correlating/associating with gender discrimination in both Spearman and Chi-Square association test. Overall, the results obtained in our study validate the results from the literature. 

The ROC curve procedure was applied to establish if the variables taken into consideration are predictive for determining the existence of gender discrimination (Table 8). This test is used to remove the zero false positives and zero false-negative situations. The area under the ROC curve quantifies the total capacity of the test to discriminate those companies that practice gender discrimination and those that are not used to gender discrimination.

The area under the curve was 0.862 (*p* = 0.034; 95%) CI (0.791 to 1), implying that the model is discriminatory for 86% of the cases. Thus, the model is discriminatory and can be utilised for measuring and identifying gender discrimination in Romania’s accounting organisations. Moreover, the test suggests that developed instruments can distinguish between normal and abnormal situations and that such a test can distinguish between them, proving that the model would correctly recognise gender discrimination and be applied to different companies with different particularities (Table 5). In conclusion, the model may correctly recognise gender discrimination. That is a confirmation of the model’s applicability in different organisations.

## 7. Discussion

Gender inequality is a reality, and Romanian society is no exception to this situation. Despite the efforts made until now and the signs of progress registered, women do not participate to the same extent as men in some of the most critical social sectors. Differences between women and men are frequent concerning activities in which they are involved, the amount of money they earn, or the responsibilities they must fulfil in their private lives [18]. 

However, the gender discrimination phenomenon is complex and subtle, difficult to quantify due to respondents’ subjectivism on cultural and sociological inheritance. The results also revealed this fact. The Spearman correlation proves that the variables are correlated with each other and not only with gender discrimination, whereas the Chi-Square test was used to test those relationships. 

The Spearman test results have shown existence related to organisational practices concerning promotion, rewards, and access to top management positions, which were considered in this study to be moderate-weakly influence factors of gender discrimination. In the meantime, logistic regression shows a moderate correlation between organisation practices and gender discrimination. That means that H1, H2, H3, and H4 are confirmed, and reinforced by the literature [3]. As the literature mentions, one can argue that gender discrimination exists because, whether on purpose or not, organisational leaders —who are generally men—are sometimes biased toward women who wish to go through the labyrinth of the organisations [38,44,54]. This study used two questions to test discrimination: one questions women concerning job access and promotion and the other regarding rewards. Women who think that men are more likely to get rewarded than women are more inclined to perceive gender discrimination in their organisations. However, it also appears that they do not see promotion, or lack thereof, as an issue or a determinant of gender discrimination. 

On the other hand, the respondents perceived the lack of networking and mentoring opportunities and high-profile tasks as gender discrimination, in line with Socratous (2018) [58]. For the Romanian context, evidence suggests that men’s networking activities have a more robust tradition and are more efficient than those organised by women following Coman (2015) and Yarram and Adapa (2022)’s results [10,24]. However, the latter are usually one-time scenarios and not activities that would ensure continuing interactions or traditions. 

Concerning individual characteristics, this study showed that married women are less likely to report gender discrimination, perhaps because they tend to incorporate their husband’s points of view [58], and thus H5 is confirmed, contrary to the results of Cohen et al. (2018) [3]. On the other hand, based on our sample, women accounting professionals with children are more likely to perceive gender discrimination in their organisations, and thus H6 is confirmed. These results are in line with the results of Anderson et al. (1994) and Claffey and Mickelson (2009) [56,61], but contrary to those of Cohen et al. (2018), who in their sample found that having children is not influencing the gender discrimination perception [3]. On the other hand, being married does not seem to influence women’s gender discrimination perceptions. 

Rank in an organisation is one factor confirmed by these respondents as influencing gender discrimination. Thus, H7 is confirmed; in a study by Bolton (2015) and Cohen et al. (2018) [3,43], women accounting professionals in higher-ranking positions were less likely to admit that there is gender discrimination in their organisations [7,59] because they may believe that such claims would invalidate their success as Chow et al. (2002). This study showed that higher-ranking women are more inclined to picture their organisations in more favourable ways than lower-ranking ones. Furthermore, individuals at the top of the hierarchical ladder are more inclined to consider the process of advancing as far as possible [35]. Lower-ranked women accounting professionals are more likely to report gender discrimination when they do not have anything to lose or any higher status to protect, as seen among those in higher organisations. A higher rank is usually linked with seniority and age, but in some cases, as Harnois (2015) mentioned, there is an interrelation between age and gender discrimination [62]. According to our results, age is not a determinant of gender discrimination, and H8 and H10 are not confirmed. 

Education level predicts that women accounting professionals who hold higher certifications are more likely to report gender discrimination. Therefore, the H9 is confirmed, and the results show that women accounting professionals who hold higher degrees are more likely to report such gender discrimination, as in the Bolton (2015) case [43]. 

In the battle between the public and private sectors (H11) [5,36], in the case of Romania, women in the private sector are more inclined to perceive that gender discrimination exists. This result confirms the results shown by Bobek et al. (2017) and Cohen et al. (2018) [3,66]. 

The correlations are all medium to weak. According to this, the determinants of gender discrimination are the existence of children, level of education, sector in which people work, promotion and rewards systems, opportunities inside organisations for networking and mentoring, and ranks occupied in the organisation. Moreover, the regression model and the factors’ *p*-values highlight the following as factors that influence gender discrimination based on the sample of respondents: rank occupied in an organisation, promotion and rewards systems, organisations’ opportunities for networking and mentoring, and the top management access, in line with the results in the relevant literature [3,5]. 

The results of the multivariate analysis of gender discrimination perceptions through investigating institutional and individual variables are consistent with the predictions. This reveals that for the sample analysed, the rewards and organisational practices influence the perception of discrimination among women accounting professionals. Concerning the intersectionality of individual characteristics interaction, our results reveal that women professionals with higher studies and children are more likely to report discrimination. In contrast, women professionals with a higher-ranking position in organisations and those working in the public sector are less likely as those from the private sector. These findings are of interest from a theoretical perspective to those who explore gender-related issues in general and in the case of accounting organisations. They are also helpful from a practical standpoint regarding the management of these accounting organisations, which should ensure gender-equitable policies for employees. 

Our results revealed the dynamic intersections [69] of institutional and individual characteristics that create a favourable context for gender discrimination in Romanian accountancy organisations. Some factors are more powerful than others, and gender discrimination results from the factors’ synergies at a specific time. 

The study conducted in Romanian accounting organisations reveals a balanced opinion concerning the existence of gender discrimination. For a deeper understanding of the results, they were analysed in correlation with (1) profession and (2) the culture, including here the traditional patriarchal mentality. 

In the Romanian accounting profession, women are well-represented in terms of numbers, yet gender discrimination persists. The traditional patriarchal mentality considers women mainly responsible for domestic chores and childcare while men are the heads of the families [1,2]. It seems that in the respondents’ view, a successful manager possesses characteristics, attitudes, and temperaments more commonly ascribed to men in general than to women in general [45]. Attitudes are less positive towards women than men leaders and potential leaders [42]. These traditional roles seem to be still perpetuated in Romanian society, causing women to either not have the ambition to pursue higher-ranking positions or forcing them to display masculine traits to be able to succeed and be taken seriously in a men’s world [18,23]. 

Other relevant explanations that might support these results refer to the cultural values of Romanians. High-power distance and the acceptance of hierarchical order without justification explain the results obtained for H1, H2, and H4 because subordinates expect to be told what to do by a benevolent autocratic manager. In addition, Romania’s collectivist culture can be considered; women would prefer to be loyal to a workgroup that values their skills. Employer/employee relationships are perceived in moral terms (like family relationships) and hiring and promotion decisions consider employees’ in-groups. In addition, interpersonal relations (manager, colleagues) are considered extremely important [26]. Beyond this, in the Romanian mentality, the focus is on ‘working to live,’ so there is no great focus on well-being, and promotion is not seen as a decisive motivational factor [26]. The high score in uncertainty avoidance is influenced by the need of maintaining rigid codes and beliefs and behaviours and it’s intolerant of unusual behaviours and ideas, which is relevant to this study. In the Romanian culture, there is an emotional need for rules (even if the rules never seem to work), time is money, people have an inner urge to be busy and work hard, precision and punctuality are the norms, innovation may be resisted, and security is an essential element in individual motivation. 

In line with Istrate (2012), respectively Faragalla and Tiron-Tudor (2020), our results revealed that this phenomenon seems to be extremely well preserved in time [16,48]. Furthermore, the study shows that women who perceive that they do not have the same opportunities, as men are more inclined to admit the existence of gender discrimination. Thus, according to our results, we can conclude that in the Romanian accounting profession context women that are perceiving the existence of different forms of gender discrimination are married with children, educated (high school or college), dedicated to their profession, and can pass through the workplace labyrinth of gender challenges. 

Our study conducted in the Romanian accounting organisations revealed a balanced opinion concerning gender discrimination but in favour of not perceiving gender discrimination in the workplace. However, for the better understanding of results and to obtain a clear overview of gender discrimination in Romanian accounting organisations, the results deserve to be analysed in correlation first with the professional context, which has become a feminised profession. Secondly, the culture, including the traditional patriarchal mentality and cultural values, plays an important role here.

## 8. Conclusions

Gender stereotypes are frequent in the Romanian context [18], and women in the profession, despite their abilities and education, still have a peripheral role in the profession’s representation. As a result, they are often affected by their negative perceptions, even though the profession is highly feminised [16]. 

Thus, the primary scope of this research was to determine which of the factors extracted from the literature are associated with gender discrimination perceptions among women accounting professionals. First, a questionnaire-based survey was conducted to collect women’s gender discrimination perceptions, and then statistical tools were employed to analyse the data. 

The associations and correlations were moderate to weak in intensity. Some of the respondents recognised that gender discrimination exists in their organisations is a victory in itself if reporting on Romanian culture in terms of mentalities. This message could be valid if speaking up and standing for one is not natural, proving that this phenomenon is hard to quantify and requires openness from respondents. Women in the private accounting sector are more inclined to experience gender discrimination but not more inclined to report it, proving the theory about an open mentality following Del Baldo et al. (2019) [26]. Women in top positions think that the pack-like behaviour of men and the boy-club attitude that causes exclusion. 

Several individual characteristics are associated with women’s perceptions concerning gender discrimination phenomena. Women with children are more likely to report gender discrimination and more likely to feel that they are being discriminated against. In addition, higher-ranking women are less likely to report gender discrimination even if they can influence women in lower-ranking positions from a mentor perspective. They are unwilling to admit that they have faced discriminatory and exclusionist behaviour might be considered a sign that organisations need to implement policies that would support feminine representation in higher ranks. 

However, the overall model and the overall answers that can change the model determine the gender discrimination variable. These results underline the intersectionality of all the variables since together; they determine whether the phenomenon exists. According to findings in the literature, it would seem that women working in the private sector are more likely to report gender discrimination than women working in the public sector [4,11,36].

One of the gaps in the literature was a comparison from an age point of view between generations regarding gender discrimination perceptions. One of this study’s findings is that older women accounting professionals (more likely to be in higher-ranking positions) are less likely to report gender discrimination in their organisations than younger ones. This result might also be a mentality or a cultural issue since Romania’s democracy is only 30 years old. Patriarchal beliefs are specific to older people, whereas more democratic views of gender are more frequent among young, unmarried people with more education, in line with the findings of Cusmir (2016) [18]. 

These results contribute to the development of knowledge in gender discrimination in an Eastern European context with a young democratic system. Moreover, the paper revealed that gender discrimination exists even in a feminised profession such as accounting. The results might be valuable insights into gender issues from managerial and political points of view. From a managerial perspective, findings claim the need to pursue several challenges. First, achieving a balanced representation within organisations is fundamental to increasing the presence of women role models and helping women aspire to leadership positions. In this regard, within extant literature, it has been widely demonstrated that mentored women are more likely to achieve the highest positions in their careers than men. Second, in a profession with high demands and time constraints, it is crucial to create a work environment that promotes gender equality by allowing workers to reconcile their work and private lives. Here there is a great need for a comprehensive strategy including a broad range of actions. We exemplify some of them: measures to reconcile family and work-life balances for both women and men, changing the political culture, overcoming gender stereotypes regarding leadership skills, ensuring women’s equal access to financial resources, and programs to support training/mentoring for women candidates to build a pipeline of women leaders. 

One of the research limitations is the fact that this study presents only the feminine perspective. However, at the same time, possible further paths for development might be to include men’s perspectives to see both sides of the story, meaning the differences in the perception of gender discrimination phenomena. Other possible developments could be at the national level to compare different professions or industries or perform comparative studies in a cross-European context. 

The final scope of this study reinforces the message that gender discrimination exists and that organisations need to do more to help prevent gender imbalance, following Haynes’s (2017) message [11].

## Figures and Tables

**Figure 1 ijerph-20-00797-f001:**
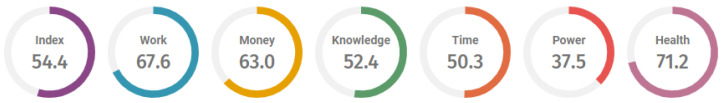
Progress on gender equality in Romania since 2010. Source: EIGE, 2020.

**Table 1 ijerph-20-00797-t001:** Institutional factors.

H	Variable	Questions	Variants of Responses	Values	Expected Sign
H1	PROMOTION	Do you believe that an employer would prefer to employ/promote a woman or a man?	Woman Man Gender does not matter	1 2 3	+
H2	REWARD	Do you believe that an employer would prefer to reward a woman or a man?	Woman Man Gender does not matter	1 2 3	+
H3	OPPORTUNITIES	Do you believe that men have more professional opportunities for development than women?	Yes No Gender does not matter	1 2 3	+
H4	TOP MANAGEMENT	Do you believe that men are preferred for top-management positions rather than women?	Woman Man Gender does not matter	1 2 3	+

**Table 2 ijerph-20-00797-t002:** Individual characteristics.

H	Variable	Variants of Responses	Values	Expected Sign
H5	MARRIED	No/Yes	1/2	−
H6	CHILDREN	No/Yes	1/2	+
H7	RANK	Low-level (entry and middle management) High-level (partners, senior executives)	1 2	−
H8	EDUCATION	no university degree university degree	1 2	+
H9	AGE	18−24 25−30 31−35 36−40 Over 40	1 2 3 4 5	+
H10	SENIORITY	0 to 5 6 to 10 11 to 20 Over 20	1 2 3 4	+
H11	SECTOR	Public Private	1 2	+/−

**Table 3 ijerph-20-00797-t003:** Demographic data (*n* = 130).

Characteristic	Number	Percentage	Characteristic	Number	Percentage
*Organisational sector*			*Age*		
Public	30	23%	18−24	7	5%
Private	100	77%	25−30	26	20%
*Organisational rank*			31−35	27	21%
High level (partners, senior executives)	57	44%	36−40	60	46%
Low level (entry, middle management)	73	56%	over 40	86	66%
*Married*			*Seniority*		
Yes	102	78.5%	0−5	11	8%
No	28	21.5%	6−10	17	13%
*Children*			11−20	35	27%
Yes	82	63%	over 20	67	52%
No	48	37%	*Gender*		
*Education*			*discrimination*		
Without university degree	8	6%	Yes	56	43%
With university degree	122	94%	No	74	57%

**Table 4 ijerph-20-00797-t004:** Gender balance at different organisational levels.

Level	Balanced	More Female	More Male
Entry	39%	50%	11%
Middle management	33%	41%	26%
Top management	29%	35%	36%

**Table 5 ijerph-20-00797-t005:** Spearman test.

Variables	AGE	MARRIED	CHILDREN	EDUCATION	SENIORITY	SECTOR	RANK	PROMOTION	REWARDS	TOP	OPPORTUNITY	GENDER DISCRIMINATION
AGE	1000	**0.192**	**0.293 ***	−0.018	**0.811 ****	**−0.285**	**−0.486 ****	0.122	0.106	0.088	−0.045	−0.023
MARRIED		1000	0.054	−0.055	**0.296 ***	0.027	**−0.174**	0.013	−0.086	−0.061	−0.163	**0.296 ***
CHILDREN			1000	0.133	**0.271 ***	−0.012	−0.117	0.010	−0.068	0.037	−0.139	**0.362 ***
EDUCATION				1000	−0.035	**0.683 ***	0.162	−0.063	−0.048	0.039	0.091	**0.305 ****
SENIORITY					1000	**−0.179**	**−0.325 ****	0.141	0.094	0.028	−0.017	−0.016
SECTOR						1000	**0.230**	−0.079	−0.060	−0.003	0.015	**0.297 ***
RANK							1000	−0.007	0.032	0.025	−0.082	0.129
PROMOTION								1000	**0.430 ****	**0.322 ****	**−0.240**	**−0.301 ***
REWARDS									1000	**0.384 ****	**−0.234**	**−0.291 ***
TOP										1000	**−0.217**	**−0.249 ***
OPPORTUNITY											1000	0.173
GENDER DISCRIMINATION												1000

Notes: * *p* < 0.05; ** *p* < 0.01 (two-tailed).

**Table 6 ijerph-20-00797-t006:** Results of the Chi-Square test of the association.

*GENDER DISCRIMINATION*	**Variable**	**Chi-Square**	***p*-Value**
	CHILDREN	5786	0.016
	EDUCATION	14	0.001
	SENIORITY	24.143	0.001
	MARRIED	34.649	0.001
	SECTOR	11.493	0.001
	RANK	22.081	0.000
	REWARDS	21.893	0.000
	PROMOTION	28.000	0.000
	OPPORTUNITIES	18.487	0.003
	TOP	10.389	0.004

**Table 7 ijerph-20-00797-t007:** Logistic regression results.

Variable	Coefficient	Std. Error	Wald	Odds Ratio	95% CI	*p*
RANK	1.932	0.747	6.685	6.9038	1.596 to 29.864	0.0097
TOP	1.142	0.512	4.993	3.1337	1.1507 to 8.5344	0.0254
OPPORTUNITY	3.544	1.739	4.149	34.5963	1.1434 to 1046.8204	0.0416
REWARD	−2.356	1.036	5.177	0.0948	0.0125 to 0.7214	0.0229
CHILDREN	−1.076	0.538	3.992	0.3411	0.1187 to 0.9798	0.0457
EDUCATION	1.436	0.671	4.585	4.2028	1.1293 to 15.6417	0.0322
SECTOR	1.108	0.657	2.844	3.0270	0.8355 to 10.9667	0.0417
Constant	−4.185	2.128	3.868			0.0492

**Table 8 ijerph-20-00797-t008:** ROC procedure.

Element	Value
Area under the ROC curve (AUC)	0.862
Standard Error	0.034
95% Confidence interval	0.791 to 1

## Data Availability

Not applicable.

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
