# Peer review of "Gender Discrimination Insights in Romanian Accounting Organisations"

_ijerph, 2023, doi:10.3390/ijerph20010797_

Round 1
Reviewer 1 Report
The article is an interesting attempt to investigate gender discrimination in accounting organizations in Romania.
The introduction is extensive and introduces the issue by referring to the historical and cultural context, which deserves praise.
The hypotheses are derived from the literature. However, I have doubts about their wording. For example, what is the meaning of hypothesis 1: Female accounting professionals who consider there is a bias in their organization with regards to selection and promotions are more like to report gender discrimination?
Is it possible to find out about gender discriminatory practices if the interviewee is convinced of unfair promotion practices in the company?
A similar problem occurs with other hypotheses. They should be rethought.
I also have doubts about the research tools used. I'm not convinced to use yes/no instead of a Likert scale. This deprives the researcher of the possibility of a more thorough examination of the relationship between the studied phenomena.
Question H3 from table 1 : Do you believe that men have more pro-fessional opportunities for developing than women?
Why is there no question of discrimination against men?
In addition, the questions are biased and suggest specific answers.
It was worth using existing and validated questionnaires.
Respondents. The idea of asking only female respondents exposes the result to bias. This is a serious weakness of the study. The explanation on page 13 (verses 608-610) is not convincing.
Reviewer 2 Report
Dear Authors,
Please find attached my review report.
Best of Luck!

Reviewer 3 Report
Dear authors:
I have reviewed the study “ Genderdiscrimination insights in the Romanian accounting organizations”. The topic is relevant and adds empirical research of gender discrimination. I was impressed for the first version of this paper. It is really well done. I only have some suggestions.
I recommended you that strength your literature review and your discussion by including Role Congruity Theory (Eagly and Karau, 2002), the paradigm Think Manager- Think Male (Schein et al. 1996) and quee bee phenomenon. I suggest you to include these theories in your article that is almost done. Congratulations. It is a pity that the sample was small. Anyway, good job.
References.
Eagly, A. H., & Karau, S. J. (2002). Role congruity theory of prejudice toward female leaders. Psychological Review, 109(3), 573–598. https://doi.org/10.1037/0033-295X.109.3.573
Schein, V. E., Mueller, R., Lituchy, T., & Liu, J. (1996). Think Manager -- Think Male: A Global Phenomenon? Journal of Organizational Behavior, 17(1), 33–41. http://www.jstor.org/stable/2488533
Round 2
Reviewer 1 Report
I appreciate the revised text, reformulation of the hypotheses and clarification of the threads in the scoreboard, as well as the extension of the research limitations.